# BIO-RFX: REFINING BIOMEDICAL EXTRACTION VIA ADVANCED RELATION CLASSIFICATION AND STRUCTURAL CONSTRAINTS

## ABSTRACT

The ever-growing biomedical publications magnify the challenge of extracting structured data from unstructured texts. This task involves two components: biomedical entity identification (Named Entity Recognition) and their interrelation determination (Relation Extraction). However, existing methods often neglect unique features of the biomedical literature, such as ambiguous entities, nested proper nouns, and overlapping relation triplets, and underutilize prior knowledge, leading to an intolerable performance decline in the biomedical domain, especially with limited annotated training data. In this paper, we propose the Biomedical Relation-First EXtraction (Bio-RFX) model by leveraging sentence-level relation classification before entity extraction to tackle entity ambiguity. Moreover, we exploit structural constraints between entities and relations to guide the model's hypothesis space, enhancing extraction performance across different training scenarios. Comprehensive experimental results on multiple biomedical datasets show that Bio-RFX achieves significant improvements on both named entity recognition and relation extraction tasks, especially under low-resource training scenarios, achieving a remarkable **5.13%** absolute improvement on average in NER, and **7.20%** absolute improvement on average in RE compared to baselines. The source code and pertinent documentation are readily accessible on established open-source repositories [1].

## 1 INTRODUCTION

Biomedical literature contains extensive knowledge and serves as a crucial medium for biomedical research. With the growing amount of biomedical publications, it has become increasingly challenging to manually keep up with the latest advances in the biomedical field. Therefore, developing methods to automatically extract structured information from unstructured biomedical texts has attracted extensive research attention. Researchers are trying to obtain biomedical entities and their relations from plain texts, namely Named Entity Recognition (NER) and Relation Extraction (RE), as illustrated in Figure 1. These structured data can be further applied to several downstream tasks and real-world circumstances in both academia and industry.

The keystone of entity and relation extraction hinges on proficiently modeling textual data, wherein it involves obtaining meaningful representations of biomedical texts and designing methods to exploit these representations. Over the past few years, tremendous success has been achieved by adapting BERT (Devlin et al., 2018) architectures to biomedical domain, including pre-training from scratch and additional training. Nevertheless, there are still two non-trivial challenges for entity and relation extraction in biomedical domain.

Firstly, learning effective representations is challenging in low-resource scenarios. Neural network-based strategies depend on substantial quantities of labeled training data, a prerequisite often elusive in the biomedical domain. This is primarily because the manual annotation of biomedical text data can be laborious, time-consuming, and error-susceptible. Annotators are required to read and interpret texts meticulously, and often reliable annotations can only be obtained from domain experts or multiple rounds of annotations on the same data.

---

[1] `https://anonymous.4open.science/r/bio-rfx-E5A9/`

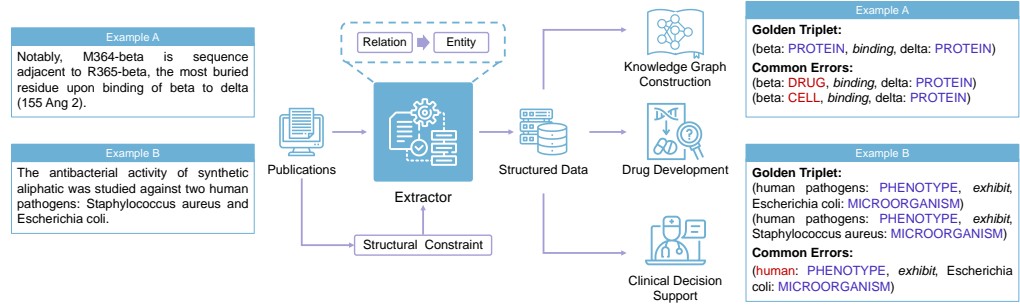

Figure 1: Automatic entity and relation extraction from biomedical publications. Example A illustrates ambiguous entities, and Example B shows perplexing nested biomedical proper nouns.

Some research concentrates on integrating biomedical knowledge graphs (KGs), such as UMLS (Bodenreider, 2004), into training data to enhance adaptability across various multiple domains (Zhang et al., 2021). Nonetheless, this approach is subject to several limitations. Entity-level KGs suffer from rapid knowledge update, large storage space, and heavy computational costs. Concept-level KGs, whose nodes and edges are abstract biomedical concepts, are affected by discrepancies in annotation standards between textual datasets and KGs. Most of the biomedical information extraction datasets center on extracting diverse and fine-grained relations between coarse-grained biomedical entities. However, concept-level KGs often fails to distinguish between various relations. For example, all the relation types in datasets DrugProt (Miranda et al., 2021) and DrugVar (Peng et al., 2017) can be categorized as the same relation type (*interact-with*) in UMLS, which severely undermine the instructive prior knowledge in KGs.

Secondly, distinct features in biomedical literature require domain-specific design for model architecture, which is a less-explored but notable aspect compared with text representations. The performance of general-domain models drops dramatically when adapting to biomedical contexts due to the stylized writing and domain-specific terminology. Moreover, biomedical entities might be ambiguous, which means the same phrases can be recognized as different named entities depending on the current context and/or the relationship with other entities. Example A in Figure 1 is a typical example, where *beta* and *delta* may refer to a variety of biomedical entities, and the relation *binding* between them hints that they are proteins. And overlapping proper nouns can be perplexing, making it difficult for the model to detect the ground truth entities. In Example B of Figure 1, *human* and *human pathogens* are both valid biomedical entities, while only the latter should be extracted under relation type *exhibit*. For these reasons, it becomes difficult for model architectures designed for general domains to effectively manage distinctive features of biomedical literature.

To address the above issues, we proposed **Bio**medical **R**elation-**F**irst EX**traction (**Bio-RFX**) model, wherein hypothesis space is constrained by prior knowledge. This architecture is inspired by the observation that there exists strong and simple structural knowledge implications among relational triplets, which cannot be ignored. For example, the relation type *exhibit* is most likely to exist between a *microorganism* and a *phenotype*. Therefore, we first predict the relation type appeared in the sentence, and then extract the relevant entities whose type satisfies such structure through a question-answering manner. A question is generated based on the relation type, and the original sentence is regarded as the context. All the related entities form a multi-span answer. Next we predict the number of valid entities in the sentence and remove the false entities with text-NMS algorithm (Hu et al., 2019). Relations between entities are generated according to the structural constraints as the final step.

This approach is capable of tackling specific issues in biomedical texts. For ambiguous entities, the predicted relation information serves as a hint for entity type. For perplexing entities, the overlapping terms are excluded by text-NMS algorithm, and thus the specificity is significantly improved.

We evaluate our method on several biomedical datasets: Bacteria Biotope (Bossy et al., 2019), DrugProt (Miranda et al., 2021), DrugVar (Peng et al., 2017). Experiment results show that our model improves the F1 scores by **5.28%** on average for NER, and by **3.70%** on average for RE task

when using the full training set. In the low resource setting, our model can obtain micro F1 gains of **5.13%** absolute improvement on average in NER, and **7.20%** absolute improvement on average in RE compared to baselines.

In summary, the main contributions of this paper include:

- We unveil an efficient biomedical relation-first extraction framework, meticulously crafted for extracting entities and relations from biomedical literature in low-resource settings.
- We construct a relation-first model to adapt to the features of biomedical texts and innovatively utilize prior knowledge to constrain the hypothesis space of the model.
- Comprehensive experiment results show that our model significantly outperforms the state-of-the-art models on multiple biomedical datasets under different settings.
- To the best of our knowledge, our work marks the inaugural endeavor in extracting both entities and relations from biomedical literature under the scenarios characterized by limited training data.

The rest of the paper is organized as follows. In Section 2, we outline the important related work. In Section 3, we introduce our proposed approach in detail. In Section 4, we present the experimental results. Finally, we conclude the paper in Section 5.

## 2 RELATED WORK

Researchers have proposed numerous methods for extracting entities and relations, most of which belong to pipeline or joint methods.

### 2.1 PIPELINE METHOD

Based upon extracting sequence, pipeline approach is divided into three paradigms.

The first paradigm starts with NER to identify all the entities in a sentence and then classifies each extracted entity pair into different relation types. To attain representations for entity and relation at various levels, FCM (Gormley et al., 2015) uses compositional embedding with hand-crafted and learned features, Zeng et al. (2014) employs convolutional deep neural network, PL-Marker (Ye et al., 2021) uses a neighborhood-oriented packing strategy and a subject-oriented packing strategy, and Fabregat et al. (2023) first trains an NER model and then transfers the weights to the triplet model. PURE (Zhong & Chen, 2020) inserts predicted entity label marks into the input sentence before relation extraction to integrate semantic information provided by entity types. Although these methods are easy to implement, they suffer from ignoring either the overlapping relation triplets or the important inner structure behind the text.

To tackle these challenges, the second paradigm is proposed. The model first detects all potential subject entities in a sentence then recognizes object entities concerning each relation. CasRel (Wei et al., 2019) regards relations as functions that map subjects to objects and identifies subjects and objects in a sequence tagging manner. Multi-turn QA (Li et al., 2019) formulates entity and relation extraction as a question-answering task, sequentially generating questions on subject entities, relations and object entities. ETL-Span (Yu et al., 2019) designs a subject extractor and a object-relation extractor and decodes the entity spans by token classification and heuristic matching algorithm. Nevertheless, in real-life circumstances, a sentence may contain a large number of entities, but relations are often sparsely distributed. Therefore, the above methods are burdened with relation redundancy. In the first paradigm, most entity pairs have no relation and in the second paradigm, enumerating all relation types is unnecessary.

The third paradigm is introduced to solve this problem, which is running relation detection at a sentence level before extracting entities. RERE (Xie et al., 2021) predicts a subset potential relations and performs a relation-specific sequence-tagging task to extract subject and object entities. PRGC (Zheng et al., 2021) adds a global correspondence used for triplet decoding. Our method Bio-RFX is different from their approaches in the following aspects. We utilize independent encoders at entity and relation extraction, which is instrumental in learning task-specific contextual representations. Besides, instead of directly applying the relation representations, we generate a

question query with respect to the relation type and targeted entity types. It provides a natural and intuitive way of jointly modeling the connection between entity and relation, and allows us to exploit the fully-fledged machine reading comprehension models. Furthermore, with a focus on domain-specific issues, typically the nested or overlapping proper nouns and biomedical terms that is difficult for recognition, we implemented text-NMS algorithm to improve specificity of extraction.

## 2.2 JOINT METHOD

Another task formulation is building joint models that simultaneously extract entities and relations. Feature-based models (Miwa & Sasaki, 2014; Ren et al., 2017) requires expertise and experience for feature engineering. Recent research focused on neural network-based models and has yielded promising results. For instance, joint extraction task can be converted to a sequence tagging problem by designing token labels that include information of entities and the relation they hold (Zheng et al., 2017). Nevertheless, these methods failed to extract overlapping entities and relation triplets, which are ubiquitous in real-world applications, especially in the biomedical domain.

To tackle the aforementioned challenge, subsequent works introduced various enhancement mechanisms via modeling input texts in a spatial rather than traditional sequential manner. TPLinker (Wang et al., 2020) regards extraction as matrix tagging instead of sequence tagging, and links token pairs with a handshake tagging scheme. OneRel (Shang et al., 2022) enumerates over all the token pairs and relations, and predicts whether they belong to any factual triplets. SPN (Sui et al., 2023) formulates joint entity and relation extraction as a direct set prediction problem. REBEL (Cabot & Navigli, 2021) takes a seq2seq approach, translating the triplets as a sequence of tokens to be decoded by the model. DeepStruct (Wang et al., 2022) pretrains language models to generate triplets from texts and performs joint entity and relation extraction in a zero-shot manner. Graph structures are also widely applied. KECI (Lai et al., 2021) first constructs an initial span-graph from the text, then uses an entity linker to form a biomedical knowledge graph. It uses attention mechanism to refine the initial span graph and the knowledge graph into a refined graph for final predictions. SpanBioER (Fei et al., 2021) is also a span-graph neural model that formulates the task as relation triplets prediction, and builds the entity-graph by enumerating possible candidate entity spans.

However, joint models have a number of drawbacks. These spatial approaches suffer from high computational complexity. Besides, NER and RE are distinct tasks, thus sharing representations between entities and relations undermines performance (Zhong & Chen, 2020). In comparison, it is much easier to divide joint extraction into several submodules and conquer each of them separately.

## 3 METHOD

In this section, we will present a detailed description of the proposed Bio-RFX, whose framework is illustrated in Figure 2. The framework contains four key components: (1) **Relation Classifier** predicts all the relation types that the input sentence expresses by performing a multi-label classification task. (2) **Entity Span Detector** extracts subject and object entities for each relation in a sentence using a relation-specific question. (3) **Entity Number Predictor** predicts the number of entities with a regression task with a question-answering approach. (4) **Pruning Algorithm** filters the candidate entities by the predicted entity number.

## 3.1 RELATION CLASSIFICATION

For relation extraction task, we detect relations at the sentence level to alleviate relation redundancy. In the example shown in Figure 2, for each relation type in the dataset, like *activator* and *inhibitor*, we will detect whether the relation is expressed in the sentence respectively. It is formulated as a multi-label classification task. Our model first constructs a contextualized representation for each input token $x_i \in x = \{x_1, x_2, ..., x_n\}$ using SciBERT (Beltagy et al., 2019). To be more specific, we construct an input sequence $[[\text{CLS}], x, [\text{SEP}]]$, feed it into the encoder and obtain the output token representation matrix $\boldsymbol{H} = [\boldsymbol{h}_0, \boldsymbol{h}_1, ..., \boldsymbol{h}_n, \boldsymbol{h}_{n+1}] \in \mathbb{R}^{d \times (n+2)}$, where $d$ indicates the hidden dimension. We then use $\boldsymbol{h}_0 \in \mathbb{R}^d$, to represent the semantic information of the sentence. Next, the sentence representation is fed into $|T_r|$ classifiers independently to determine whether the sentence expresses relation $\tau_r$, where $\tau_r \in T_r$. For relation $\tau_r$, the output of the classifier $\hat{p}_r$ can be defined by $\hat{p}_r = \sigma(\boldsymbol{W}_r \boldsymbol{h}_0 + \boldsymbol{b}_r)$, where $\boldsymbol{W}_r, \boldsymbol{b}_r$ are trainable model parameters and denote the weight and

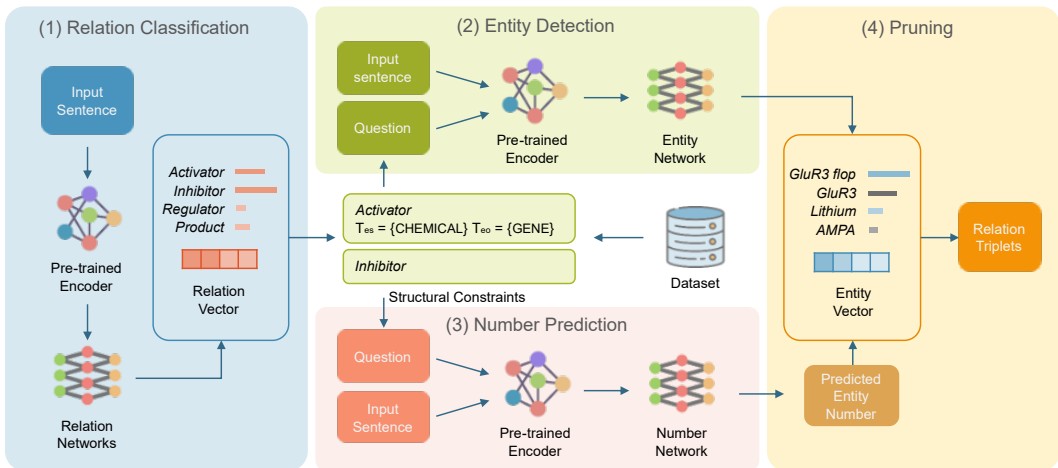

Figure 2: The overall framework of Bio-RFX. (1) The relation classifier predicts that there are two relations in the sentence, *Activator* and *Inhibitor*. (2–4) Relation-specific entity extraction is performed for each of the predicted relation type. To be more specific, (2) the entity detector extracts all the entities that satisfy the structural constraints via a question-answering manner, and (3) the number predictor outputs the number of spans similarly. (4) The relation triplets are generated by excluding the overlapping perplexing entities.

bias respectively. $\sigma$ is the sigmoid activation function. For each relation $\tau_r$, we employ the cross entropy loss to optimize the training process. Let $p_r$ denote the ground truth from annotated data; $p_r = 1$ is used to represent that relation $\tau_r$ has appeared in the sentence and vice versa. Therefore, the loss function for relation classifier can be defined as:

$$\mathcal{L}_{\mathrm{rel}} = -\sum_{x \in D} \sum_{r=1}^{|T_r|} p_r \log \hat{p}_r. \tag{1}$$

## 3.2 ENTITY EXTRACTION

### 3.2.1 ENTITY DETECTION

We formulate entity detection as span extraction from the sentence. This approach is inspired by machine reading comprehension models that extract answer spans from the context. For the first step, we design a question for entity detection. For NER, we generate a question $q$ using predefined templates with all the entity types in $T_e$. For example, if $T_e = \{null, chemical, gene, variant\}$, then $q = $ *What are the chemicals, genes and variants in the sentence?* RE is more complicated, since the strong structural constraints between entity types and relation types should not be ignored. For RE, the question is specific for each relation type $\tau_r$ appeared in the sentence. Given a relation type $\tau_r$, let $T_{es}$ denote the set of allowed subject entity types, and $T_{eo}$ denote the set of allowed object types. We obtain these two sets by enumerating all the possible triplets in the dataset as prior knowledge, which is undemanding since the relation types are fine-grained while the entity types are coarse-grained, resulting in limited size of $T_{es}$ and $T_{eo}$. Suppose $\tau_r = activator$, then $T_{es} = \{chemical\}$, $T_{eo} = \{gene\}$. The question is generated with $T_{es} \cup T_{eo}$, i.e. $q_r = $ *What gene does the chemical activate?* Note that it is a relatively simple approach. For other prompting techniques, please refer to Appendix A. Given the question, we regard the sentence $x$ as context and build the input sequence $[[\mathrm{CLS}], q_r, [\mathrm{SEP}], x, [\mathrm{SEP}]]$. Then, we compute the representation of each span $s \in S$ in sentence $x$. Let $\boldsymbol{H} = [\boldsymbol{h}_1, \boldsymbol{h}_2, \ldots, \boldsymbol{h}_N]$ be the token representation matrix for the input sequence, where $N$ denotes the number of tokens in the sequence. We obtain the representation $\boldsymbol{s}$ for $s$ using an attention mechanism over tokens (Lee et al., 2017):

$$a_t = \frac{\exp\left(\mathrm{FFNN}_\alpha(s_t^*)\right)}{\sum_{k=l_S}^{l_E} \exp\left(\mathrm{FFNN}_\alpha(s_k^*)\right)}, \tag{2}$$

$$\boldsymbol{s} = [\boldsymbol{h}_{l_S}, \sum_{t=l_S}^{l_E} a_t \boldsymbol{h}_t, \boldsymbol{h}_{l_E}, \Phi(w)], \qquad (3)$$

where $s^*$ denotes the concatenation of all the tokens in the span $s$; weight $a_t$ denotes the normalized attention score; $l_S, l_E$ denote the start and end position for span $s$ respectively; and $\Phi(w)$ is a learnable width embedding for the span width $w = l_E - l_S$. Then, we compute the probability $\hat{p}_e$ that span $s$ is an entity with type $\tau_e$ using a FFNN with GELU activation function, namely $\hat{p}_e = \text{FFNN}_e(\boldsymbol{s})$.

The loss function for NER during training is defined in the following equation:

$$\mathcal{L}_{\text{ent}} = -\sum_{x \in D} \sum_{s \in S_x} \sum_{e=0}^{|T_e|} w_e p_e \log \hat{p}_e. \qquad (4)$$

For RE, the input sequence is relation-specific, thus the loss function is:

$$\mathcal{L}_{\text{ent}} = -\sum_{x \in D} \sum_{s \in S_x} \sum_{e=0}^{|T_e|} \sum_{\substack{r=1 \\ \tau_r \in R_x}}^{|T_r|} w_e p_{re} \log \hat{p}_{re}. \qquad (5)$$

In both cases, we use $w_e$ to handle the overwhelming negative entity labels, namely *null* entity when $e = 0$, with a focus on valid entities. Specifically, we decrease the weight for *null* label to 0.1 in the cross entropy loss function.

### 3.2.2 NUMBER PREDICTION

In order to exclude perplexing entities from the output, we implement textual non-maximum suppression (text-NMS) algorithm (Hu et al., 2019), which requires us to predict the number of potential entities in a sentence $x$. We formulate the regression task in a question-answering manner. In the above example, for NER, we have $q = $ *How many chemicals, genes and variants are there in the sentence?* For RE, for each subject-object pair $\langle \tau_{es}, \tau_{eo} \rangle \in T_{es} \times T_{eo}$, a unique question is generated. For instance, $\tau_r = $ *activator*, $\tau_{es} = $ *chemical*, and $\tau_{eo} = $ *gene*, then $q_r = $ *How many chemicals and genes are there in the sentence with relation activation?* The question and the sentence are concatenated together using [CLS] and [SEP] to form the input sequence. Similar to Section 3.1, we obtain the representation vector $\boldsymbol{h}_0$ for the input sequence and then utilize a feed forward neural network (FFNN) with GELU activation function to acquire the predicted number $\hat{k}$ of potential entities, namely $\hat{k} = \text{FFNN}_n(\boldsymbol{h}_0)$.

We use $k$ to denote the number of ground truth entities in a sentence. The loss function for number prediction in NER is the mean squared loss, which can be defined as:

$$\mathcal{L}_{\text{num}} = \sum_{x \in D} (k - \hat{k})^2. \qquad (6)$$

For RE, it is slightly different concerning relations. We define $k_r$ as the number of subjects and objects with relation $\tau_r$, and duplicate entities are only counted once. The loss is defined as:

$$\mathcal{L}_{\text{num}} = \sum_{x \in D} \sum_{\substack{r=1 \\ \tau_r \in R_x}}^{|T_r|} (k_r - \hat{k}_r)^2. \qquad (7)$$

### 3.2.3 PRUNING ALGORITHM

After spans are extracted, we then adopt the text-NMS algorithm to heuristically prune redundant and perplexing entities. Firstly, for each span $s$, we obtain the confidence score $\lambda(s) = 1 - \hat{p}_{e=0}$, namely the probability for not being a *null* entity. Then the spans in $S$ are sorted according to the descending order of their confidence scores. A new set $\hat{S}$ is initialized as the final prediction for spans. We select the span $s_i$ that possesses the highest confidence score, put $s_i$ into $\hat{S}$, remove any remaining span $s_j \in S$ that overlaps with $s_i$ from $S$, and remove $s_i$ from $S$ as well. Text-level F1 score is used to indicate the degree of overlapping. This process is repeated until either $|\hat{S}|$ reaches

$k$, i.e. the number of entities, or $S$ becomes an empty set. A detailed illustration for the algorithm is described in Algorithm 1 in Appendix B.

We then generate relation triplets with the spans in $\hat{S}$. Instead of adopting a nearest-matching method (Xie et al., 2021), we match all the possible subjects and objects to address the overlapping triplets in biomedical texts. To be more specific, for relation $\tau_r$, each $\langle \tau_{es}, \tau_{eo} \rangle \in T_{es} \times T_{eo}$ is converted to a relation triplet $\langle \tau_{es}, \tau_r, \tau_{eo} \rangle$ as the final result.

# 4 EXPERIMENTS AND ANALYSIS

In this section, we conduct extensive experiments on sentence-level NER and RE to justify the effectiveness of the model. We first introduce the datasets in use and experimental settings. The performance evaluation and analysis of the models are presented. Furthermore, we examine our method in a low-resource scenario. Finally, we provide the results for ablation study to demonstrate the effects of each submodule in the proposed framework.

## 4.1 DATASETS

We empirically study and evaluate related methods on three datasets: Bacteria Biotope (Bossy et al., 2019), DrugProt (Miranda et al., 2021), DrugVar (Peng et al., 2017). For more details and preprocessing methods, please refer to Appendix C.

## 4.2 EXPERIMENTAL SETTINGS

We outline the baselines and metrics here. For implementation details, please refer to Appendix D.

### 4.2.1 BASELINES

We evaluate our model by comparing with several models that are capable of both entity and relation extraction on the same datasets, which are strong models designed for general domain (PURE (Zhong & Chen, 2020), TPLinker-plus (Wang et al., 2020)) and for biomedical domain (KECI (Lai et al., 2021) and SpanBioER (Fei et al., 2021)). Some of the competitive relation-first approaches, such as PRGC (Zheng et al., 2021), use ground truth entities as input, making them unsuitable for baseline models due to their lack of NER application.

### 4.2.2 EVALUATION METRIC

We use micro F1 scores for both NER and RE to evaluate models. An entity is considered matched if the whole span and entity type match the ground truth, and a relation triplet is regarded correct if the relation type and both subject entity and object entity are all correct.

## 4.3 MAIN RESULTS

Table 1: Micro F1 (%) of models on biomedical datasets. The best results are in bold.

| Model | Bacteria Biotope | | DrugVar | | DrugProt | |
|---|---|---|---|---|---|---|
| | NER | RE | NER | RE | NER | RE |
| PURE | 66.35 | 38.16 | 80.48 | 65.20 | 90.74 | 70.07 |
| KECI | 63.95 | 37.29 | 73.93 | 63.17 | 85.71 | **74.19** |
| TPLinker-plus | 69.25 | 37.49 | 79.54 | 63.14 | 91.03 | 70.61 |
| SpanBioER | 73.60 | 38.30 | 81.39 | 67.56 | 88.28 | 65.94 |
| Bio-RFX | **75.90** | **43.38** | **84.12** | **69.28** | **91.87** | 71.22 |

Table 1 shows the micro F1 scores of all the models on the three datasets. The results demonstrate that our model improves the F1 scores by 5.28% on average for NER for all datasets, and by 3.70%

on average for RE. On Bacteria Biotope and DrugVar, our model significantly outperforms all the other baselines in both tasks, please refer to Appendix E for detailed significance tests. The powerful model KECI achieves competitive performance in RE on DrugProt, but it performs poorly in NER on the same dataset. KECI models information extraction with graphical structure, thus it is able to generate more accurate relation triplets compared to our simple generating method. However, its training process depends heavily on a large amount of annotated data, leading to unsatisfactory results on smaller datasets. In contrast, on a more practical biomedical dataset with insufficient annotated training data, the proposed method performs better than other baseline models.

We can draw several conclusions from the observations. Firstly, our method achieves superior performance compared baselines for biomedical datasets, especially those with limited training data, which indicates that individual encoder can effectively learn precise representations for biomedical texts. In addition, in datasets that has annotation discrepancies with knowledge bases and therefore is challenging to conduct entity linking, strong structural constraints in biomedical domain can indeed help achieve better performance than traditional methods that fuse KGs into the model.

## 4.4 Low Resource Setting

Table 2: Micro F1 (%) of models on biomedical datasets under low resource setting. The best results are in bold. The number in the bracket indicates the approximate size of training set.

| Model | DrugVar (500) | | DrugVar (200) | | DrugProt (500) | | DrugProt (200) | |
|---|---|---|---|---|---|---|---|---|
| | NER | RE | NER | RE | NER | RE | NER | RE |
| PURE | 76.88 | 58.37 | 73.55 | 50.01 | 89.48 | 57.14 | 83.43 | 51.53 |
| KECI | 75.03 | 59.35 | 66.94 | 52.86 | 82.26 | 52.46 | 67.18 | 46.58 |
| TPLinker-plus | 77.32 | 60.66 | 71.06 | 23.53 | 85.95 | 46.29 | 78.11 | 32.80 |
| SpanBioER | 77.56 | 60.12 | 74.13 | 50.00 | 87.13 | 49.25 | 84.78 | 44.00 |
| Bio-RFX | **81.23** | **63.90** | **74.90** | **54.01** | **89.67** | **57.58** | **87.43** | **52.03** |

We conduct several experiments to explore the effectiveness of our method in a low resource scenario. We randomly select 10% and 4% samples from DrugProt, as well as 50% and 20% samples from DrugVar to construct new datasets. The experiment results are shown in Table 2. Bio-RFX consistently outperforms baseline methods, achieving 5.13% average improvement in NER and 7.20% average improvement in RE across all datasets.

Compared with pipeline and joint methods, our model excels in the following aspects: (1) Number prediction effectively improves specificity by pruning perplexing entities in biomedical domain. Therefore, our approach performs better than models that simply adopt span extraction for entity and relation extraction task (such as PURE). (2) Dividing complicated task into several submodules significantly decreases the difficulty of training. Joint methods which based on intricate tagging scheme severely suffer from scarce training data. For instance, TPLinker-plus combines information from the whole triplet and the whole span to construct labels for span pair, resulting in 4 variants for each relation type. Hence, the $4|T_r|$-class classification task contributes to great learning difficulty and significant performance drop under the low resource setting. On the contrary, our divide-and-conquer philosophy is more effective because task-specific representation helps to achieve better performance. (3) KG-enhanced joint methods are affected by overwhelming noisy prior knowledge from KGs when training data is limited. In biomedical datasets, the definition for *null* entity varies greatly, as specific entities (for example, qualitative concepts such as *revealed* or *active*) are likely to be considered as *null* entity if not the primary focus of the dataset. Comprehensive KGs recognize these entities incorrectly when training samples only account for a small portion of the total input data. To support this argument, we find that KECI has lower precision and higher recall across the experiments, while our model shows the opposite. Using extensive knowledge base as prior knowledge in low resource scenario leads to overfitting to KGs, and constraining the hypothesis space of the model is a more preferable alternative.

Additionally, the Large Language Models (LLMs) have shown promise in few-shot natural language processing tasks. In Appendix G, we demonstrate the comparative results between Bio-RFX and GPT-4.

## 4.5 Ablation Study

Table 3: Ablation study on biomedical datasets.

| Model | Bacteria Biotope | | DrugVar | | DrugProt | |
|---|---|---|---|---|---|---|
| | NER | RE | NER | RE | NER | RE |
| Bio-RFX | **75.90** | **43.38** | 84.12 | 69.28 | **91.87** | **71.22** |
| Bio-RFX (- Structure) | 75.90 | 20.80 | 84.12 | 36.31 | 91.87 | 27.97 |
| Bio-RFX (- Number) | 72.87 | 42.30 | **84.68** | **70.82** | 90.74 | 65.51 |

Table 4: Ablation study on biomedical datasets under low resource setting.

| Model | DrugVar (500) | | DrugVar (200) | | DrugProt (500) | | DrugProt (200) | |
|---|---|---|---|---|---|---|---|---|
| | NER | RE | NER | RE | NER | RE | NER | RE |
| Bio-RFX | **81.23** | **63.90** | **74.90** | **54.01** | **89.67** | **57.58** | 87.43 | **52.03** |
| Bio-RFX (- Structure) | 81.23 | 30.98 | 74.90 | 28.34 | 89.67 | 22.83 | 87.43 | 22.19 |
| Bio-RFX (- Number) | 80.40 | 61.86 | 74.13 | 51.85 | 89.43 | 56.94 | **89.35** | 43.24 |

In Table 3–4, we show the micro F1 score of our model without structural constraints for relation triplet generation (- Structure), without entity span number prediction (- Number), and the full model on the datasets. For Bio-RFX (- Structure), instead of enumerating each $\langle \tau_{es}, \tau_{eo} \rangle \in T_{es} \times T_{eo}$ for relation $\tau_r$ to produce relation triplets, we regard each entity pair in $T_{ev} \times T_{ev}$ as a subject-object pair for relation $\tau_r$, where $T_{ev}$ indicates the set of valid and not-*null* entities. Since structural constraints only influences relation triplet generation, the result for NER remains the same with the full model. For Bio-RFX (- Number), we remove the number prediction and use the average number of entities in a sentence as the threshold for text-NMS algorithm during inference. Without number prediction, the micro F1 scores for NER and RE drop by 0.50% and 2.70% on average, respectively. The results indicate the model's performance is promoted with the presence of both structural constraints and number prediction, of which strong structural constraints between entity types and relation types are most helpful. It proves the ability of our model to tackle perplexing entities and take advantage of structural constraints of relation triplets in biomedical literature.

## 4.6 Case Study

To assess the model's comprehension of ambiguous biomedical entities, we study several typical cases. The results are presented in Appendix F.

## 5 Conclusion and Future Work

This paper introduces Bio-RFX, a novel biomedical entity and relation extraction method, using structural constraints for relation triplets to constrain the hypothesis space. The model can address ambiguous entities and relation redundancy using a relation-first extraction approach, and uses a heuristic pruning algorithm to recognize complex overlapping entity spans more precisely. It overcomes annotated training data limitations, and significantly improves entity and relation extraction performance. Extensive experimental results on multiple real-world biomedical datasets with abundant and limited training data show that our method significantly outperforms the state-of-the-art methods on NER and RE. For future work, we will explore the following research directions: (1) We will expand the capability of the proposed method via incorporating other knowledge representation methods besides structural constraints obtained by statistic features. (2) We will explore more effective ways to generate questions or hints for relation-specific tasks, so as to make better use of the rich semantic information provided in pre-trained encoders. (3) We will address the error propagation issue for pipeline training, which may lead to a discrepancy between training and testing.

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

## A  PROMPT TECHNIQUES

We have explored the following prompt techniques. However, incorporating these prompt modules has negatively impacted the model's performance. In contrast, our designed question template turned out to be more effective.

## A.1 Term Definitions

We enrich the question with definitions of types of entities and relations to provide the model with semantic information in the biomedical domain. For instance, the relation-specific question *What gene does the chemical activate?* is followed by the definition of activator obtained from the Free Medical Dictionary[2], i.e., *An activator is a substance that makes another substance active or reactive, induces a chemical reaction, or combines with an enzyme to increase its catalytic activity*. The results are shown in Table 5 and Table 6, i.e. Bio-RFX (+definition). It can be observed that the Micro F1 scores for NER and RE decrease. We believe the contextualized knowledge representation during the pre-training process is sufficient, and the rigid definitions merely introduce noise to data distribution.

Table 5: Micro F1 (%) of models on biomedical datasets.

| Model | Bacteria Biotope | | DrugVar | | DrugProt | |
|---|---|---|---|---|---|---|
| | NER | RE | NER | RE | NER | RE |
| Bio-RFX | 75.90 | 43.38 | 84.12 | 69.28 | 91.87 | 71.22 |
| Bio-RFX (+definition) | 75.14 | 44.65 | 83.86 | 69.70 | 90.79 | 56.79 |

Table 6: Micro F1 (%) of models on biomedical datasets under low resource setting. The number in the bracket indicates the approximate size of the training set.

| Model | DrugVar (500) | | DrugVar (200) | | DrugProt (500) | | DrugProt (200) | |
|---|---|---|---|---|---|---|---|---|
| | NER | RE | NER | RE | NER | RE | NER | RE |
| Bio-RFX | 81.23 | 63.90 | 74.90 | 54.01 | 89.67 | 57.58 | 87.43 | 52.03 |
| Bio-RFX (+definition) | 79.90 | 63.22 | 71.19 | 48.68 | 88.67 | 52.50 | 89.35 | 56.73 |

## A.2 UMLS Markers

External biomedical knowledge is also considered when designing prompts. We use UMLS Metamap, a handy toolkit based on a biomedical knowledge graph, to match the biomedical terms in the text and insert unique markers both before and after the terms. Take the following sentence as an example.

Some clinical evidences suggested that pindolol can be effective at producing a shortened time to onset of antidepressant activity.

In this sentence, *pindolol* is recognized by Metamap as a pharmacologic substance. When type-specific markers are used, the result is:

> Some clinical evidences suggested that **<DRUG>** pindolol **</DRUG>** can be effective at producing a shortened time to onset of antidepressant activity.

The type of the entity can also be masked, i.e.,

> Some clinical evidences suggested that **<ENTITY>** pindolol **</ENTITY>** can be effective at producing a shortened time to onset of antidepressant activity.

On the DrugProt dataset, we observed a 3.02% and 6.45% decrease in micro F1 scores for NER and RE, respectively. Several reasons may contribute to this experience results. To begin with, the entity types in Metamap and the entity types in the datasets are quite different, posing a challenge for entity linking. Another reason is that the matching method is mainly based on the syntax tree and searching, thus the matching accuracy is not satisfactory. In the following example, the term

---

[2]https://medical-dictionary.thefreedictionary.com/

'of' is erroneously identified as a gene (OF (TAF1 wt Allele)) due to its ambiguous nature, which subsequently hampers the overall performance. Moreover, Metamap extracts all the entities without being conscious of the relation type expressed in the sentence, misleading our entity model.

> ... **<CHEMICAL>** isoprenaline **</CHEMICAL>** - induced maximal relaxation ( E ( max ) ) **<GENE> of </GENE> <CHEMICAL>** methacholine **</CHEMICAL>** - contracted preparations in a concentration dependent fashion ...

## B  TEXTUAL NMS ALGORITHM

A detailed description for the algorithm is presented in Algorithm 1.

---

**Algorithm 1** Textual Non-Maximum Suppression

---

**Require:** spans $S$, span number threshold $k$;
**Ensure:** pruned spans $\hat{S}$;
  Sort $S$ in descending order of span scores;
  $\hat{S} = \{\}$;
  **while** $S \neq \{\}$ and $|\hat{S}| < k$ **do**
    **for** $s_i$ in $S$ **do**
      $\hat{S} = \hat{S} \cup \{s_i\}$;
      $S = S - \{s_i\}$;
      **for** $s_j$ in $S$ **do**
        **if** $F1(s_i, s_j) > 0$ **then**
          $S = S - \{s_j\}$;
        **end if**
      **end for**
    **end for**
  **end while**

---

## C  DATASETS AND PREPROCESSING

We will briefly review all the datasets below and state the preprocessing methods we have applied. The statistics of the datasets are listed in Table 7.

1. **Bacteria Biotope** is a part of the BioNLP Open Shared Tasks[3]. We knuckle down to the entity and relation extraction subtask which aims to recognizing mentions of microorganisms and microbial biotopes and phenotypes in scientific and textbook text, and extracting relations between them. Manually annotated data is provided in the dataset. The original dataset is designed for document-level information extraction, which is beyond the scope of this paper. Thus we split the texts into sentences with Punkt (Kiss & Strunk, 2006) sentence tokenizer and ignore all the cross-sentence relation triplets. The test set is not publicly available, so we present the experiment results on the validation set hereafter for fair comparison.

2. **DrugVar** is a subset of N-ARY dataset proposed in Peng et al. (2017) and mainly focuses on extracting fine-grained interactions between drugs and variants. The dataset was constructed by first obtaining biomedical literature from PubMed Central[4] and then identifying entities and relations with distant supervision from Gene Drug Knowledge Database (Dienstmann et al., 2015) and Clinical Interpretations of Variants In Cancer[5] knowledge bases. It is also designed for document-level information extraction, so we adopt the aforementioned method for sentence segmentation during preprocessing.

---

[3]http://2019.bionlp-ost.org
[4]http://www.ncbi.nlm.nih.gov/pmc/
[5]http://civic.genome.wustl.edu/

3. **DrugProt** is a track in BioCreative VII and focuses on extracting a variety of important associations between drugs and genes/proteins to understand gene regulatory and pharmacological mechanisms. The data is collected from PubMed abstracts and then manually labeled by domain experts. We also perform sentence segmentation during preprocessing. We as well merge some of the relation types so that all the refined relation labels are at the same level in the relation concept hierarchy.

Table 7: Statistics of datasets

| Datasets | #Ent Type | #Rel Type | #Train | #Valid |
|---|---|---|---|---|
| Bacteria Biotope | 6 | 3 | 284 | 153 |
| DrugVar | 3 | 4 | 929 | 267 |
| DrugProt | 3 | 6 | 6,273 | 1,377 |

# D  IMPLEMENTATION DETAILS

For fair comparison, all the models use *scibert-scivocab-cased* (Beltagy et al., 2019) as pre-trained Transformer encoder. We consider spans with up to L = 8 words, which covers 97.89% of the entities on average in the datasets. We train our models with Adam (Kingma & Ba, 2014) optimizer of a linear scheduler with a warmup ratio of 0.1. We train the relation, entity and count model for 100 epochs, and a learning rate of 1e-5 and a batch size of 8. We use gold relations and entity numbers to train the entity model and the predicted relations and numbers during inference. To be more specific, for each relation, if the probability obtained by the relation classifier is above the relation specific threshold, then the sentence will be classified as positive, which means the sentence is expressing this relation. Otherwise, it will be classified as negative. The relation-specific threshold can be optimized via maximizing the classification F1 score on the validation set.

# E  SIGNIFICANCE TESTS

We designed a statistical analysis and performed experiments, which we address as follows.

Table 8: Significance tests on DrugVar.

| Run | Bio-RFX | | PURE | | KECI | | TPLinker-plus | | SpanBioER | |
|---|---|---|---|---|---|---|---|---|---|---|
| | NER | RE | NER | RE | NER | RE | NER | RE | NER | RE |
| 1 | 82.55 | 68.65 | 80.24 | 63.40 | 74.28 | 63.34 | 79.54 | 63.14 | 82.18 | 69.39 |
| 2 | 82.73 | 69.37 | 81.18 | 66.42 | 74.25 | 63.40 | 80.13 | 61.61 | 81.90 | 68.39 |
| 3 | 83.06 | 71.57 | 80.52 | 65.53 | 74.98 | 60.00 | 78.88 | 61.60 | 81.75 | 68.01 |
| 4 | 83.85 | 70.45 | 80.58 | 64.90 | 74.30 | 63.00 | 78.76 | 61.16 | 81.44 | 67.48 |
| 5 | 83.63 | 70.30 | 80.42 | 66.03 | 74.96 | 65.08 | 82.03 | 67.33 | 81.84 | 67.76 |
| $t$ | – | – | 8.13 | 8.78 | 33.76 | 5.99 | 5.40 | 5.52 | 3.76 | 2.39 |
| $p$ | – | – | 0.0006 | 0.0005 | 0.0000 | 0.0020 | 0.0028 | 0.0026 | 0.0099 | 0.0375 |

1. We choose 5 seeds randomly.

2. We train Bio-RFX and all the baseline models with each seed and record the corresponding performances.

3. We perform one-tailed paired t-tests between Bio-RFX and each baseline model with significance level $\alpha = 0.05$ on the results. For each baseline model:

   (a) We compute the difference in performance between Bio-RFX and the baseline model so that we obtain 5 difference measures $d_i$ $(i = 1, 2, \ldots, 5)$.

(b) We compute the $t$ statistic under the null hypothesis that Bio-RFX and the compared baseline have equal performance:

$$t = \frac{\bar{d} - 0}{s/\sqrt{5}} = \frac{\sqrt{5}\bar{d}}{\sqrt{\frac{1}{4}\sum_{i=1}^{5}(d_i - \bar{d})^2}},$$

where $\bar{d}$ and $s$ are the sample mean and standard deviation of the difference measures, respectively.

(c) We compute the p-value and compare it to the significance level $\alpha = 0.05$. If the p-value is smaller than $0.05$ or the $t$ statistic is bigger than $2.132$, we reject the null hypothesis.

The $t$ statistics and p-values between Bio-RFX and the baseline models are shown in Table 8, 9 and 10. We can observe that all the p-values are below $\alpha = 0.05$ (and all the $t$ statistics are above $2.132$), rejecting the null hypothesis and demonstrating that Bio-RFX significantly outperforms the all the baselines under general setting, and most of the baselines when training resources are limited.

Table 9: Significance tests on Bacteria Biotope.

| Run | Bio-RFX | | PURE | | KECI | | TPLinker-plus | | SpanBioER | |
|-----|---------|------|------|------|------|------|------|------|------|------|
| | NER | RE | NER | RE | NER | RE | NER | RE | NER | RE |
| 1 | 75.94 | 45.83 | 66.59 | 36.47 | 66.20 | 36.36 | 69.25 | 37.49 | 74.78 | 44.68 |
| 2 | 76.65 | 42.55 | 66.59 | 36.18 | 62.02 | 32.14 | 69.73 | 44.58 | 74.94 | 42.32 |
| 3 | 75.84 | 43.77 | 65.80 | 37.99 | 63.83 | 31.03 | 69.15 | 41.55 | 75.18 | 41.90 |
| 4 | 76.25 | 46.19 | 66.04 | 36.75 | 59.57 | 38.71 | 70.88 | 38.35 | 75.65 | 43.42 |
| 5 | 76.29 | 45.39 | 67.38 | 37.24 | 68.15 | 34.48 | 66.77 | 40.33 | 74.64 | 42.92 |
| $t$ | – | – | 38.84 | 10.37 | 7.86 | 11.81 | 10.38 | 2.23 | 4.92 | 3.69 |
| $p$ | – | – | 0.0000 | 0.0002 | 0.0007 | 0.0001 | 0.0002 | 0.0448 | 0.0040 | 0.0105 |

Table 10: Significance tests on DrugProt(200).

| Run | Bio-RFX | | PURE | | KECI | | TPLinker-plus | | SpanBioER | |
|-----|---------|------|------|------|------|------|------|------|------|------|
| | NER | RE | NER | RE | NER | RE | NER | RE | NER | RE |
| 1 | 88.12 | 55.63 | 83.90 | 55.74 | 71.05 | 38.58 | 78.11 | 32.80 | 82.24 | 41.31 |
| 2 | 88.55 | 53.97 | 83.40 | 51.41 | 68.57 | 38.43 | 79.79 | 26.52 | 81.87 | 39.65 |
| 3 | 88.97 | 57.04 | 84.29 | 52.89 | 72.76 | 38.59 | 84.70 | 25.00 | 82.17 | 42.92 |
| 4 | 88.93 | 55.59 | 84.13 | 54.62 | 74.64 | 35.58 | 82.75 | 26.29 | 82.03 | 42.06 |
| 5 | 90.49 | 58.78 | 84.09 | 58.23 | 71.07 | 44.17 | 82.68 | 30.23 | 82.38 | 42.01 |
| $t$ | – | – | 13.69 | 2.11 | 16.61 | 17.61 | 7.37 | 18.62 | 19.22 | 26.16 |
| $p$ | – | – | 0.0001 | 0.0512 | 0.0000 | 0.0000 | 0.0009 | 0.0000 | 0.0000 | 0.0000 |

# F  CASE STUDY

Figure 3 illustrates cases of ambiguous entities in DrugProt dataset. In case A, *Abeta* is a chemical in the form of a peptide, as well as processed from the Amyloid precursor protein. In case B, *angiotensin II* is both a medication used to increase blood pressure, and a type of protein. Since DrugProt focuses on extracting drug-gene/protein interactions, both of them are considered to be proteins in the context. With the structural constraints, our model is able to correctly predict the ground truth labels.

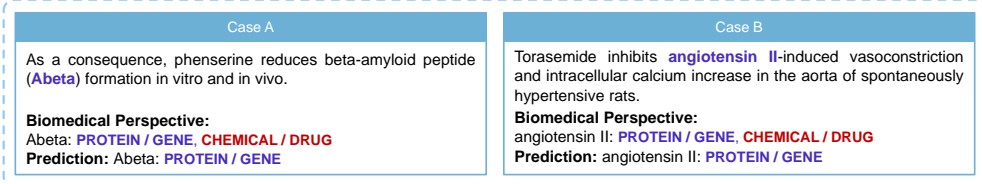

Figure 3: Case study for ambiguous biomedical entities.

## G  POTENTIAL OF LLMS

With the rapid development of Large Language Models (LLMs), it is necessary to discuss the potential of LLMs for our task. We choose GPT-4 (OpenAI, 2023) to jointly conduct NER and RE on biomedical texts.

To inform GPT-4 about its role and our task, we first send a system message, i.e. *You are stepping into the role of an expert assistant specialized in biomedicine. Your primary task is to accurately extract entities and relations from biomedical texts and respond to users' queries with clear, concise, and precise answers.*

After the system message, we give several examples. Each example contains a question section and an answer section. A question section consists of 4 parts:

1. The biomedical text where we extract entities and relations.
2. The entity and relation types specified by the dataset.
3. The structural constraints between the entity and relation types.
4. A question guiding GPT-4 to provide the answer.

An answer section consists of 2 parts:

1. The entities detected from the text. To facilitate entity extraction, we inform GPT-4 to generate highly structurized answers, e.g. `<BCRP|GENE>` represents an entity *BCRP* of type GENE. In practice, we perform Chain of Thought (Wei et al., 2022) prompting to enhance accuracy.
2. The relation triplets extracted from the text. Similar to entity detection, GPT-4 intends to generate structurized answers, e.g. `<Menthol|CHEMICAL|TRPM8|GENE|activator>` represents an *activator* relation, whose subject and object are *Menthol* and *TRPM8*.

Finally, we form a question section based on the biomedical text and send it to GPT-4. We perform regular expression matching on the response message to retrieve the answers. The evaluation metrics are consistent with the previous sections, i.e. an entity is considered matched if the whole span and entity type match the ground truth, and a relation triplet is regarded correct if the relation type and both subject entity and object entity are all correct.

Table 11: Micro F1 (%) of GPT-4 and Bio-RFX on biomedical datasets.

| Model | Bacteria Biotope | | DrugVar (500) | | DrugVar (200) | | DrugProt (500) | | DrugProt (200) | |
|---|---|---|---|---|---|---|---|---|---|---|
| | NER | RE | NER | RE | NER | RE | NER | RE | NER | RE |
| GPT-4 | 53.76 | 29.07 | 61.86 | 12.62 | 61.97 | 6.94 | 67.29 | 26.25 | 69.80 | 32.26 |
| Bio-RFX | **75.90** | **43.38** | **81.23** | **63.90** | **74.90** | **54.01** | **89.67** | **57.58** | **87.43** | **52.03** |

Results are shown in table 11. For DrugVar and DrugProt, the results of GPT-4 are approximately the same as their subsets, since the given examples are the same. Compared to Bio-RFX, GPT-4

underperforms severely in all experimental settings, indicating that although LLMs have numerous emergent abilities, biomedical RE remains a difficult problem for LLMs. The source code is publicly available at `https://anonymous.4open.science/r/bio-re-gpt-F0A9/`.

