# OpenReview forum: "Bio-RFX: Refining Biomedical Extraction via Advanced Relation Classification and Structural Constraints"
_ICLR.cc/2024/Conference — Submitted to ICLR 2024_

### Official Review · Reviewer_apnY · 2023-10-25

**Soundness:** 3 good
**Presentation:** 3 good
**Contribution:** 3 good
**Rating:** 6
**Confidence:** 4

**Summary:**

This paper proposes a novel method for entity relation extraction from text, where relation types are detected first, and then entities (or arguments) are detected later. For entities, the entity candidates and their number for each relation type are detected using two types of question-answering framework, and the entities are extracted by filtering the entity candidates considering the number of entities. The approach shows the best performance among compared methods for the named entity recognition and relation extraction tasks on three biomedical data sets in the full data and low resource settings, except for relation extraction on one data set in the full data set setting. Ablation studies show the usefulness of the structure constraints and number prediction.

**Strengths:**

- The approach to first detect relation types and then entities is novel
- The paper is well-written and easy to follow. Figure 2 is helpful to grasp the overall framework.
- The results on three datasets show high performance in both full data and low resource settings, and the ablation study shows the usefulness of the proposed enhancements.

**Weaknesses:**

- The approach could be generally applied to other domains, but the approach is presented as a model for the biomedical domain, and the scope is limited.
- The comparison with existing state-of-the-art entity relation models is limited. The authors presented several approaches like OneRel and SPN in the related work section, and there are several SOTA models for entity relation tasks as follows, but the comparison is not performed.
  - Pere-Lluís Huguet Cabot and Roberto Navigli. 2021. REBEL: Relation Extraction By End-to-end Language generation. In Findings of the Association for Computational Linguistics: EMNLP 2021, pages 2370–2381, Punta Cana, Dominican Republic. Association for Computational Linguistics.
  - Chenguang Wang, Xiao Liu, Zui Chen, Haoyun Hong, Jie Tang, and Dawn Song. 2022. DeepStruct: Pretraining of Language Models for Structure Prediction. In Findings of the Association for Computational Linguistics: ACL 2022, pages 803–823, Dublin, Ireland. Association for Computational Linguistics.
  - Deming Ye, Yankai Lin, Peng Li, and Maosong Sun. 2022. Packed Levitated Marker for Entity and Relation Extraction. In Proceedings of the 60th Annual Meeting of the Association for Computational Linguistics (Volume 1: Long Papers), pages 4904–4917, Dublin, Ireland. Association for Computational Linguistics.

**Questions:**

- Please see the weaknesses above.
- How is the model specific to the biomedical domain?
- Appendix A shows several comparisons of prompting using domain resources, but the prompts used in practice are not clear. For the activator relation type, the authors use the prompt including activate (not activator), so is question generation done manually? They also say, "Note that it is a relatively simple approach" in explaining questions, but do they use any other complicated approach in practice? Or is this simple approach the best?

---

> ### Author Response · Authors · 2023-11-18
> **Rebuttal - part 1**
>
> We appreciate your constructive feedback, which has been instrumental in enhancing our paper. We have addressed all the points raised and have detailed our responses below.
>
> **Weaknesses**
>
> > The approach could be generally applied to other domains, but the approach is presented as a model for the biomedical domain, and the scope is limited.
>
> This approach is designed to tackle specific issues in biomedical texts. The relation-first approach is based on the high prevalence of ambiguous terms in biomedical literature and serves as a hint for the entity type. The implementation of structural constraints is informed by domain-specific knowledge. For instance, the relation *activator* can only occur between a *chemical* and a *gene*. On the other hand, the relation *located_in* can occur between *sports_team* and *city*, *company* and *country*, *city* and *country*, among others, thus offering only a very weak constraint. We will explore the potential of applying our approach to other domains in our future research.
>
> > The comparison with existing state-of-the-art entity relation models is limited. The authors presented several approaches like OneRel and SPN in the related work section, and there are several SOTA models for entity relation tasks as follows, but the comparison is not performed.
>
> Inspired by your constructive comments, we have updated the related work section in the revised version of the paper. We will further conduct comprehensive research into these SOTA models, extending them into biomedical fields and performing in-depth comparisons and discussions.

---

> ### Author Response · Authors · 2023-11-18
> **Rebuttal - part 2**
>
> **Questions**
>
> > Please see the weaknesses above.
>
> Kindly refer to our reply to Weakness 1 and Weakness 2.
>
> > How is the model specific to the biomedical domain?
>
> Please review our response to Weakness 1.
>
> > Appendix A shows several comparisons of prompting using domain resources, but the prompts used in practice are not clear. For the activator relation type, the authors use the prompt including activate (not activator), so is question generation done manually? They also say, "Note that it is a relatively simple approach" in explaining questions, but do they use any other complicated approach in practice? Or is this simple approach the best?
>
> In practice, we exclusively use the questions and refrain from using any prompts derived from domain resources.
>
> All the questions are generated based on templates except for entity detection (Section 3.2.1) in relation extraction task. The primary reason is that the relation types in different datasets are morphologically and semantically diverse, and manual generation is more accurate in terms of syntax and semantics. In our future work, we plan to investigate techniques for automatically generating questions using the appropriate toolkit. While for the other components in our model, the question is generated based on templates. For example, in number prediction, the template is *How many $\tau_{e1}, \tau_{e2}, ..., \tau_{eN}$ are there in the sentence with relation $\tau_r$?*, where $\tau_{e1}, \tau_{e2}, ..., \tau_{eN}$ are all the entity types that satisfy the structural constraint of relation $\tau_r$.
>
> In Appendix A we also mentioned two prompting techniques: term definitions and UMLS markers. Both of them have a negative influence on the model's performance.
>
> **Term Definition.** We enrich the question with definitions from the Free Medical Dictionary (https://medical-dictionary.thefreedictionary.com/), i.e. Bio-RFX (+definition). In NER, the question is followed by the definitions of all the entity types that appear in the question. Similarly, for RE, definitions of the present relation type. The experimental results are displayed below. The rigid definitions introduce noise into the data distribution, thereby increasing the complexity of modeling sentence representations.
>
> | Dataset          | Task | Bio-RFX | Bio-RFX (+definition) |
> | :--------------- | :--- | :------ | :-------------------- |
> | DrugProt         | NER  | 91.87   | 90.79                 |
> |                  | RE   | 71.22   | 56.79                 |
> | DrugProt(500)    | NER  | 89.67   | 88.67                 |
> |                  | RE   | 57.58   | 52.50                 |
> | DrugProt(200)    | NER  | 87.43   | 89.35                 |
> |                  | RE   | 52.03   | 56.73                 |
> | DrugVar          | NER  | 84.12   | 83.86                 |
> |                  | RE   | 69.28   | 69.70                 |
> | DrugVar(500)     | NER  | 81.23   | 79.90                 |
> |                  | RE   | 63.90   | 63.22                 |
> | DrugVar(200)     | NER  | 74.90   | 71.19                 |
> |                  | RE   | 54.01   | 48.68                 |
> | Bacteria Biotope | NER  | 75.90   | 75.14                 |
> |                  | RE   | 43.38   | 44.65                 |
>
> **UMLS Marker.** As suggested in Appendix A, incorporating UMLS markers leads to a performance drop, due to the discrepancies between entity types of UMLS Metamap and datasets, the low accuracy of Metamap matching, and the overlooking of relation types in the sentence. In the following example, the term *of* is erroneously identified as a gene *OF (TAF1 wt Allele)* due to its ambiguous nature, which subsequently hampers the overall performance. Moreover, accessing Metamap via web API is extremely time-consuming, posing a challenge to processing speed.
>
> > ... **<CHEMICAL>** isoprenaline **</CHEMICAL>** - induced maximal relaxation ( E ( max ) ) **<GENE>** of **</GENE>** **<CHEMICAL>** methacholine **</CHEMICAL>** - contracted preparations in a concentration dependent fashion ...

---

> > ### Comment · Reviewer_apnY · 2023-11-23
> >
> > Thank you for the response and update. The response is not enough to raise my score, so I will keep it.

---

### Official Review · Reviewer_Wvyh · 2023-10-29

**Soundness:** 2 fair
**Presentation:** 2 fair
**Contribution:** 3 good
**Rating:** 5
**Confidence:** 3

**Summary:**

This paper studies the problem of information extraction (named entity recognition and relation extraction) in the biomedical domain.
The authors propose to predict the relation type in the sentence and then extract the relevant entities in a question-answering manner. Finally, the pruning algorithm is used with an entity number predictor to filter the final predicted entities.
The proposed method is evaluated on three biomedical datasets: Bacteria Biotope, DrugProt, and DrugVar, and results show that the proposed method outperforms several baselines, especially under the low-resource scenario.

**Strengths:**

* The authors propose an interesting paradigm for relation extraction: predict relation first and then extract entities; extract entities in a QA manner.
* The authors report strong results of the proposed method on several biomedical datasets.

**Weaknesses:**

* The choice of baselines seems arbitrary; I suggest linking Section 2.1 and Section 4.2.1 to make the reason for ‘why cannot use relation-first baseline’ more explicit.
* It isn't easy to gain insights into the main strengths of the proposed method. See question A

**Questions:**

Question A: it is unclear what the ablated variant ‘- structure’ is. Suggest linking Section 4.5 and the four components in Section 3. For example, the ‘- Number’ variant contains only component 1, 2, 4? and, it would be nice to see a variant containing only the first two components

---

> ### Author Response · Authors · 2023-11-18
> **Rebuttal**
>
> We appreciate your insightful suggestions and apologize for any lack of clarity in our paper. We have addressed all the issues, and the detailed responses are as follows.
>
> **Weaknesses**
>
> > The choice of baselines seems arbitrary; I suggest linking Section 2.1 and Section 4.2.1 to make the reason for "why cannot use relation-first baseline" more explicit.
>
> In Section 2.1, we introduce two relation-first methods: PRGC and RERE, both of which are not suitable baselines for our task.
>
> PRGC only performs the relation classification task, which means that it uses ground truth entities as the input and predicts the relations between entity pairs. On the contrary, our method is able to extract both entities and relation triplets.
>
> RERE extracts subjects and objects from the text, ignoring entities that have not been found in any relation triplets, while our approach can recognize all the entities in the text. Besides, RERE does not differentiate between multiple mentions of the same entity within the sentence. However, biomedical literature is seethed with complicated clauses and ambiguous terms, which necessitates the precise identification of each unique mention. Thus, the benchmark is designed to distinguish each mention during the process of metric calculation, and RERE has to be excluded from baselines.
>
> > It isn't easy to gain insights into the main strengths of the proposed method. See question A.
>
> Please refer to our response to question A.
>
> **Questions**
>
> > It is unclear what the ablated variant "- Structure" is. Suggest linking Section 4.5 and the four components in Section 3. For example, the "- Number" variant contains only component 1, 2, 4? and, it would be nice to see a variant containing only the first two components.
>
> Referring to Section 3, our framework contains four key components:
>
> 1. Relation Classifier
> 2. Entity Span Detector
> 3. Entity Number Predictor
> 4. Pruning Algorithm
>
> Apart from the four key components mentioned above, we take advantage of the structural constraints brought by domain knowledge to obtain more accurate relation triplets.
>
> In addition to the brief explanation on page 9 of our revised paper, we present a more detailed discussion here. In order to measure the significance of the structural constraints, we performed the ablated variant "- Structure" by removing the structural constraints informed by domain-specific knowledge from the model. Without taking advantage of the structural constraints, the model will end up obtaining less accurate relation triplets. For example, the relation *activator* can only occur between a *chemical* and a *gene*. The ablated variant without the structural constraints may extract relation triplets containing relation *activator* and two *gene*'s.
>
> In order to test the validity of the entity number predictor, we performed the ablated variant "- Number" containing only components 1, 2, and 4, for which we use the average number of entities in a sentence as the threshold for the pruning algorithm during inference.
>
> We have performed an ablated variant containing only the first two components, i.e. the relation classifier and the entity span detector. While the micro F1 score for NER on the DrugProt dataset increased by 1.35%, the micro F1 score for RE on the same dataset decreased dramatically by 4.58%. This demonstrates that although the ablated variant may recall slightly more entities, it will extract a lot more false relation triplets due to the existence of perplexing entities, thus harming the precision.

---

> > ### Comment · Reviewer_Wvyh · 2023-11-22
> > **acknowledgement**
> >
> > I have read the author's response and other reviews. My concern regarding the choice baselines has not been addressed.

---

> ### Author Response · Authors · 2023-11-23
>
> Thank you for your prompt feedback. As outlined in our previous response under **Weakness 1**, a fair comparison between Bio-RFX and relation-first baselines is not feasible. We appreciate your understanding and look forward to further discussions on this matter.

---

### Official Review · Reviewer_FKtM · 2023-10-31

**Soundness:** 3 good
**Presentation:** 3 good
**Contribution:** 3 good
**Rating:** 5
**Confidence:** 4

**Summary:**

The paper introduced a novel biomedical entity and relation extraction method that deploys structural constraints for relation triplets to constrain the hypothesis space. It reported on extensive evaluations on three datasets and in a case study to provide convincing evidence of performance gains obtained using the introduced method. It supplemented these performance evaluations by conduction an ablation study as well.

**Strengths:**

The paper introduced a novel biomedical entity and relation extraction method that deploys structural constraints for relation triplets to constrain the hypothesis space. Biomedical applications of this kind are of substantial societal importance.

It reported on extensive evaluations on three datasets and in a case study to provide convincing evidence of performance gains obtained using the introduced method. It supplemented these performance evaluations by conduction an ablation study as well.

The paper was very carefully written. It was clear and convincing.

**Weaknesses:**

I was unable to find information about statistical analysis (e.g., statistical significance tests or confidence intervals).

Automatic entity and relation extraction is a trending topic in research on natural language processing, knowledge graphs, and machine/deep learning. Hence, a more convincing case for novel contributions made in this paper could be made. For example, I could not find a single paper on contrastive representation learning included in the paper, although, e.g., triplet loss and contrastive representation learning are very closely related (see, e.g., Le-Khac, P. H., Healy, G., & Smeaton, A. F. (2020). Contrastive representation learning: A framework and review. IEEE Access, 8, 193907-193934).

Reference list of the paper could be perfected and math should be punctuated.

**Questions:**

How were the performance gained evaluated as significant? Were they both statistically and practically significant?

What made the contributions made new/novel compared to prior work?

---

> ### Author Response · Authors · 2023-11-18
> **Rebuttal**
>
> Thank you for the constructive comments. They are very helpful for improving our paper. We understand your concerns and would like to address them as follows.
>
> **Weaknesses**
>
> > I was unable to find information about statistical analysis (e.g., statistical significance tests or confidence intervals).
>
> Thank you for your constructive suggestions. We are conducting more experiments regarding statistical tests according to your new comment and will update the results before November 22.
>
> > Automatic entity and relation extraction is a trending topic in research on natural language processing, knowledge graphs, and machine/deep learning. Hence, a more convincing case for novel contributions made in this paper could be made. For example, I could not find a single paper on contrastive representation learning included in the paper, although, e.g., triplet loss and contrastive representation learning are very closely related (see, e.g., Le-Khac, P. H., Healy, G., & Smeaton, A. F. (2020). Contrastive representation learning: A framework and review. IEEE Access, 8, 193907-193934).
>
> Thank you for your comments, which have broadened our perspective. We will conduct in-depth research and discuss it in detail in our subsequent work.
>
> > Reference list of the paper could be perfected and math should be punctuated.
>
> Thank you for your detailed and helpful suggestions. We have carefully polished the paper and fixed the writing issues in the revised manuscript.
>
> **Questions**
>
> > How were the performance gained evaluated as significant? Were they both statistically and practically significant?
>
> Please see our response to Weakness 1.
>
> > What made the contributions made new/novel compared to prior work?
>
> Compared with other methods, our approach introduces novelty in the following aspects. Firstly, we take advantage of the structural constraints brought by domain knowledge to obtain more accurate relation triplets. This approach mitigates the issue of extracting false positive triplets from the texts when enumerating all the entity pairs. Secondly, in order to address domain-specific issues, such as nested or overlapping biomedical terms, we implemented the text-NMS algorithm to improve the specificity of extraction. Thirdly, we generate a question query with respect to the relation type and targeted entity type, providing an intuitive way of jointly modeling the connection between entity and relation.

---

> ### Comment · Reviewer_FKtM · 2023-11-20
> **Reviewer response to authors' rebuttal**
>
> It would have been helpful to communicate the results from those additional statistical tests in the rebuttal already. Because these results - or even methodological detail towards conducting the tests - were not included, I am not in a position of changing my review for the better. Based on the other reviewers' concerns, I have now revised the review to go from marginally above to marginally below the acceptation threshold.

---

> ### Author Response · Authors · 2023-11-20
> **Statistical analysis designs and results - part 1**
>
> Thank you for your prompt response. We have designed a statistical analysis and performed experiments on the DrugVar dataset according to your constructive suggestions, which we address as follows.
>
> 1. We choose 5 seeds randomly.
>
> 2. We train Bio-RFX and all the baseline models with each seed and record the corresponding performances.
>
> 3. We perform one-tailed paired t-tests between Bio-RFX and each baseline model with significance level $\alpha = 0.05$ on the results. For each baseline model:
>
>    1. We compute the difference in performance between Bio-RFX and the baseline model so that we obtain 5 difference measures $d_i~(i = 1, 2, \dots, 5)$.
>
>    2. We compute the $t$ statistic under the null hypothesis that Bio-RFX and the compared baseline have equal performance:
>       $$
>                   t = \frac{\bar{d} - 0}{s / \sqrt{5}} = \frac{\sqrt{5}\bar{d}}{\sqrt{\frac14\sum_{i=1}^5(d_i - \bar{d})^2}},
>       $$
>       ​        where $\bar{d}$ and $s$ are the sample mean and standard deviation of the difference measures, respectively.
>
>    3. We compute the p-value and compare it to the significance level $\alpha = 0.05$. If the p-value is smaller than $0.05$ or the $t$ statistic is bigger than $2.132$, we reject the null hypothesis.
>
> The $t$ statistics and p-values between Bio-RFX and the baseline models are shown in the following table. We can observe that all the p-values are below $\alpha = 0.05$ (and all the $t$ statistics are above $2.132$), rejecting the null hypothesis and demonstrating that Bio-RFX significantly outperforms all the compared baselines. We hope these results may alleviate your concerns. We have updated the experimental results, please refer to our latest response.
>
> | Run  | Bio-RFX |       | PURE   |        | KECI   |        | TPLinker-plus |        | SpanBioER |        |
> | ---- | ------- | ----- | ------ | ------ | ------ | ------ | ------------- | ------ | --------- | ------ |
> |      | NER     | RE    | NER    | RE     | NER    | RE     | NER           | RE     | NER       | RE     |
> | 1    | 82.55   | 68.65 | 80.24  | 63.40  | 74.28  | 63.34  | 79.54         | 63.14  | 82.18     | 69.39  |
> | 2    | 82.73   | 69.37 | 81.18  | 66.42  | 74.25  | 63.40  | 80.13         | 61.61  | 81.90     | 68.39  |
> | 3    | 83.06   | 71.57 | 80.52  | 65.53  | 74.98  | 60.00  | 78.88         | 61.60  | 81.75     | 68.01  |
> | 4    | 83.85   | 70.45 | 80.58  | 64.90  | 74.30  | 63.00  | 78.76         | 61.16  | 81.44     | 67.48  |
> | 5    | 83.63   | 70.30 | 80.42  | 66.03  | 74.96  | 65.08  | 82.03         | 67.33  | 81.84     | 67.76  |
> | $t$  | $-$     | $-$   | 8.13   | 8.78   | 33.76  | 5.99   | 5.40          | 5.52   | 3.76      | 2.39   |
> | $p$  | $-$     | $-$   | 0.0006 | 0.0005 | 0.0000 | 0.0020 | 0.0028        | 0.0026 | 0.0099    | 0.0375 |

---

> ### Author Response · Authors · 2023-11-22
> **Statistical analysis designs and results - part 2**
>
> We have further conducted significance tests using the Bacteria Biotope dataset, and we present here our experimental results on the Bacteria Biotope dataset. It shows that Bio-RFX significantly outperforms the baselines on this dataset too. Due to our limited computing resources and time constraints, we may not be able to conduct the significance tests for the DrugProt dataset before Nov. 22, as it is a much bigger dataset, instead, we will see if we can perform the experiments for the DrugProt dataset in the low resource setting (i.e., with much smaller training data) by the due date. In future versions, we will carry out more comprehensive significance tests and analyses. We appreciate your understanding and patience.
>
> | Run  | Bio-RFX |       | PURE   |        | KECI   |        | TPLinker-plus |        | SpanBioER |        |
> | ---- | ------- | ----- | ------ | ------ | ------ | ------ | ------------- | ------ | --------- | ------ |
> |      | NER     | RE    | NER    | RE     | NER    | RE     | NER           | RE     | NER       | RE     |
> | 1    | 75.94   | 45.83 | 66.59  | 36.47  | 66.20  | 36.36  | 69.25         | 37.49  | 74.78     | 44.68  |
> | 2    | 76.65   | 42.55 | 66.59  | 36.18  | 62.02  | 32.14  | 69.73         | 44.58  | 74.94     | 42.32  |
> | 3    | 75.84   | 43.77 | 65.80  | 37.99  | 63.83  | 31.03  | 69.15         | 41.55  | 75.18     | 41.90  |
> | 4    | 76.25   | 46.19 | 66.04  | 36.75  | 59.57  | 38.71  | 70.88         | 38.35  | 75.65     | 43.42  |
> | 5    | 76.29   | 45.39 | 67.38  | 37.24  | 68.15  | 34.48  | 66.77         | 40.33  | 74.64     | 42.92  |
> | $t$  | $-$     | $-$   | 38.84  | 10.37  | 7.86   | 11.81  | 10.38         | 2.23   | 4.92      | 3.69   |
> | $p$  | $-$     | $-$   | 0.0000 | 0.0002 | 0.0007 | 0.0001 | 0.0002        | 0.0448 | 0.0040    | 0.0105 |

---

> ### Author Response · Authors · 2023-11-23
> **Statistical analysis designs and results - part 3**
>
> We have conducted significance analysis for the DrugProt dataset in the low resource setting (i.e., with a much smaller training set of size 200). The results are shown below. We would like to highlight that even when training resources are limited, our method is still able to demonstrate significant performance improvements over the most of the baselines. We appreciate your time and consideration, and we look forward to any further comments or suggestions you may have to enhance our work.
> | Run  | Bio-RFX |       | PURE   |        | KECI   |        | TPLinker-plus |        | SpanBioER |        |
> | :--- | :------ | :---- | :----- | :----- | :----- | :----- | :------------ | :----- | :-------- | :----- |
> |      | NER     | RE    | NER    | RE     | NER    | RE     | NER           | RE     | NER       | RE     |
> | 1    | 88.12   | 55.63 | 83.90  | 55.74  | 71.05  | 38.58  | 78.11         | 32.80  | 82.24     | 41.31  |
> | 2    | 88.55   | 53.97 | 83.40  | 51.41  | 68.57  | 38.43  | 79.79         | 26.52  | 81.87     | 39.65  |
> | 3    | 88.97   | 57.04 | 84.29  | 52.89  | 72.76  | 38.59  | 84.70         | 25.00  | 82.17     | 42.92  |
> | 4    | 88.93   | 55.59 | 84.13  | 54.62  | 74.64  | 35.58  | 82.75         | 26.29  | 82.03     | 42.06  |
> | 5    | 90.49   | 58.78 | 84.09  | 58.23  | 71.07  | 44.17  | 82.68         | 30.23  | 82.38     | 42.01  |
> | $t$  | $-$      | $-$    | 13.69  | 2.11   | 16.61  | 17.61  | 7.37          | 18.62  | 19.22     | 26.16  |
> | $p$  | $-$     | $-$    | 0.0001 | 0.0512 | 0.0000 | 0.0000 | 0.0009        | 0.0000 | 0.0000    | 0.0000 |

---

### Official Review · Reviewer_wMCa · 2023-11-01

**Soundness:** 3 good
**Presentation:** 3 good
**Contribution:** 2 fair
**Rating:** 6
**Confidence:** 4

**Summary:**

The paper tackles the issue of Named Entity Recognition and Biomedical Relation Extraction. In particular, it proposes a framework that starts with identifying relations first, and using that information to constrain the space for entity extraction. The paper shows that the method surpasses SOA for NER and RE tasks on multiple biomedical datasets, on a variety of scenarios, including low-resource.

**Strengths:**

- **Quality**: Approach is grounded in biological knowledge of constraining the space for entity extractions to types of relationship
- **Originality**: The paper combines various ML concepts in an interesting and creative way
- **Results**: Results show improvement over other SOA models

**Weaknesses:**

The concepts introduced in the paper are not novel. However, the paper does a nice job at putting them in a creative way to obtain improvements over SOA.

**Questions:**

- Looking at the results in Table 1: What is the hypothesis for KECI performing better on RE task in DrugProt?

---

> ### Author Response · Authors · 2023-11-18
> **Rebuttal**
>
> Thank you for the valuable and detailed comments on the experimental results, which are very helpful for improving our paper.
>
> **Questions**
>
> > Looking at the results in Table 1: What is the hypothesis for KECI performing better on RE task in DrugProt?
>
> In addition to the brief explanation on the last line of page 7, we present a more detailed discussion here. The result in Table 1 can be attributed to two key factors: the integration of a background Knowledge Graph (KG) within KECI, and the large number of training samples in the DrugProt dataset.
>
> KECI first constructs an initial span-graph from the text and then uses an entity linker to form a knowledge graph containing relevant background biomedical knowledge for the entity mentions in the text. To make the final predictions, KECI fuses the initial span graph and the knowledge graph into a more refined graph using an attention mechanism. The intricate network of interconnections among entities within the background KG equips KECI with a deeper understanding of sentence structures, thereby facilitating the formation of complex triplets. On the other hand, in order to lessen the demand for computational resources, we match all the possible subjects and objects to generate triplets, as stated in Section 3.2.3.
>
> Another contributing factor is the larger size of the DrugProt training set compared to other datasets, as detailed in Table 5 in Appendix C. This allows KECI to effectively strike a balance between utilizing training data and incorporating prior knowledge. However, when training data is limited, it will lead to overfitting to the background KG. For additional supporting experimental results and discussions, please refer to Section 4.4.

---

### Author Response · Authors · 2023-11-22

We would like to express our gratitude to all the reviewers for the time and effort they spent reviewing our paper. We value the insightful and constructive comments, which have been instrumental in improving our work. We have carefully addressed all the comments in our individual responses. We kindly invite further engagement to resolve any confusion and to receive feedback on our rebuttal.

In the following, we list the changes in the revised manuscript.
* Supplemented Appendix A with experimental results of incorporating domain knowledge into prompts.
* Added significant analysis in Appendix E.
* Punctuated mathematical formula in Section 3, as suggested by Reviewer FKtM.
* Elucidated the setting of ablation study in Section 4.5.
* Added missing related works in Section 2, in response to Reviewer FKtM and Reviewer apnY.

---

### Meta-Review · Area_Chair_iosE · 2023-12-06

**Metareview:**

This work considered biomedical named entity recognition and relation extraction via a staged approach which sequentially predicts relation types, then entity identification (followed by pruning). The intuition is to bake the constraints implicit in relation extraction into the method.

The approach is relatively straightforward (a strength!), and seem to offer gains on the biomedical datasets considered. However, the baselines considered are weak (as pointed out by Wvyh and apnY). In particular, the authors compare against methods only from 2021 and before; this is a long time in NLP years. Indeed, as pointed out by apnY, generative methods for this task have since emerged as SOTA (e.g., Cabot and Navigli [EMNLP 2021], Wang et al [ACL 2022], Ye et al. [ACL 2022], Wadwha et al. [ACL 2023]). Without comparison to at least one such generative baseline, it is difficult to appreciate the contribution here. The authors addressed this in response only elaborating their "related work" section, but provide no additional empirical results.

**Justification For Why Not Higher Score:**

The main limitation here is the lack of modern baselines; given that this is a very applied and therefore empirical paper (the primary contribution being a simple, intuitive technique for biomedical RE), this is critical in my view.

**Justification For Why Not Lower Score:**

N/A

---

### Decision · Program_Chairs · 2024-01-16

Reject